# HYBRID POLICIES USING INVERSE REWARDS FOR RE-INFORCEMENT LEARNING

## ABSTRACT

This paper puts forward a broad-spectrum improvement for reinforcement learning algorithms, which combines the policies using original rewards and inverse (negative) rewards. The policies using inverse rewards are competitive with the original policies, and help the original policies correct their mis-actions. We have proved the convergence of the inverse policies. The experiments for some games in OpenAI gym show that the hybrid polices based on deep Q-learning, double Q-learning, and on-policy actor-critic obtain the rewards up to 63.8%, 97.8%, and 54.7% more than the original algorithms. The improved polices are more stable than the original policies as well.

## 1 INTRODUCTION

### 1.1 MOTIVATION

Reinforcement learning, as one of the most crucial branch in machine learning area, has now been broadly studied and improved thus become more and more powerful and accurate since DeepMind introduced deep Q-learning Mnih et al. (2013). Recently, a couple of successors of deep Q-learning have been developed and applied on many areas van Hasselt et al. (2016) Wang et al. (2016). Meanwhile, the classical actor-critic algorithms Konda & Tsitsiklis (2000) played ideas from SARSA Rummery & Niranjan (1994) and deep Q-learning as well.

In a traditional mathematical setting, suppose that we want to approximate the target stationary value function $Q^*(s, a)$ with a fixed basis $v_i$, so that the result function,

$$Q(s, a; \theta) = \sum_i \theta_i v_i(s, a), \tag{1}$$

minimizes the loss $(Q^*(s, a) - Q(s, a; \theta))^2$.

Then the gradient descent optimization algorithm will help us finding the minimum of the above by repeatedly using the following updating formula,

$$\theta_{t+1} = \theta_t + \alpha(Q^* - Q(s_t, a_t; \theta_t))\nabla_{\theta_t} Q(s_t, a_t; \theta_t). \tag{2}$$

In Q-learning, one of the main challenge is that the target stationary value function $Q^*(s, a)$ in equation 2 cannot be computed directly. To resolve this problem people needs to estimate $Q^*(s, a)$ in equation 2. For example, in $Q$ learning the following estimation function is used,

$$Q^*(s_t, a_t) \approx Y_t^Q = R_t + \gamma \max_{b \in \mathbf{A}} Q(s_{t+1}, b; \theta_t). \tag{3}$$

$Y_t^Q$ is also called *target* for the $Q^*$ value. With the different target network parameter $\theta'$, deep Q-learning uses,

$$Q^*(s_t, a_t) \approx Y_t^{DQN} = R_t + \gamma \max_{b \in \mathbf{A}} Q(s_{t+1}, b; \theta_t'). \tag{4}$$

It is well understood that deep Q-learning has intrinsic overestimations Thrun & Schwartz (1993) and the neural network introduces more significant error in the computation of the iterations van Hasselt et al. (2016). Although double Q-learning van Hasselt et al. (2016) and its successors claimed some improvements, the effectiveness is not good enough.

We confirmed that the estimation error is not neglectable and will occasionally leads to fluctuation. For example, we tried deep Q-learning, double Q-learning, and on-policy actor-critic with neural network for CartPole Barto et al. (1983) in OpenAI gym ope (a). (See figure 1) .

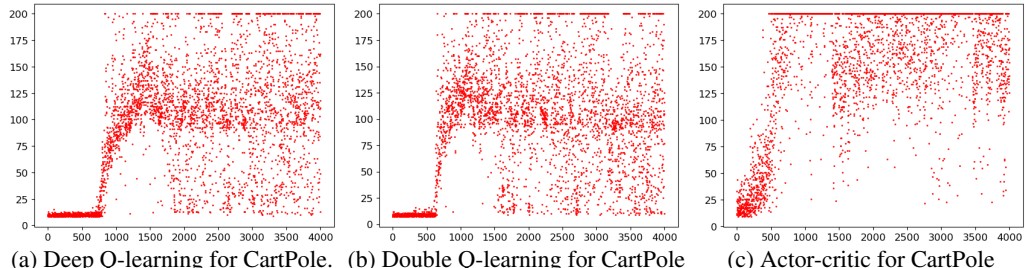

(a) Deep Q-learning for CartPole.  (b) Double Q-learning for CartPole  (c) Actor-critic for CartPole

Figure 1: RL algorithms for CartPole. x-axis is the iterations of episodes and y-axis is the rewards of the corresponding episode. The goal of the game is to keep the pole standing on the cart as long as possible. The maximum number of steps , i.e., the sum of the rewards, of each episode is 200.

Recall that the environment of CartPole returns 1 reward point for each step until the pole is down or the cart is out of range. The player needs to push the cart to the left or the right to keep the pole not falling down and the cart in the range as long as possible, i.e., getting more reward points.

The experiment shows the unstability of these reinforcement learning algorithms. In figure 1a, the policy is degraded significantly and is quite unstable even after it continuously gets 200 points. So do double Q-learning (Figure 1b) and actor-critic (figure 1c). All experiments in figure 1 show that the policies degrade dramatically even after they reach the goal for a while. We cannot see the trend of the convergence in 4000 episodes. The unstability means that more training may not result in better policies. Even worse, the users of these algorithms may be afraid of more training and do not know when to stop learning.

Based on these observation, we believe that it will be helpful to find a metric to test whether the estimated $Q^*$ ($Y^Q$ or $Y^{DQN}$) we used in $Q$-learning is too "risky". For such purpose, we introduced *inverse policy* to measure the badness of a state and uses it to fix $Y^Q$, $Y^{DQN}$, or other targets.

We believe that a good state should not be too bad in the measure of inverse policy which measures the badness of a state. In addition to use the optimistic estimation $Y^Q$ or other targets, we also think pessimistically and use inverse policy to find and fixes those "risky" estimation in equation 1.

Once the above tuition has been established, we successfully developed our hybrid policy and showed that the overall performance and stability of RL algorithms can be improved dramatically. Also we believe that this motivation can be further developed, see further works in section 5.

## 1.2 OUR APPROACHES

In the traditional reinforcement learning, the agent gets rewards from the environment and uses the rewards as the feedback in the algorithm. For a particular state, a traditional reinforcement learning algorithm tries finding the action that leads to the maximum rewards in the future.

Let us change to the opposite perspective. What if we find the worst action for a particular state? For CartPole, if we find the worst action, the other action must be the best. Hence, the alternative approach is to learn the "worst" actions in this case. For more complicated action spaces, without loss of generality, we can find out both the maximum and the minimum values because any value in $Q(s, a; \theta_t)$ is known in value function-based algorithms. If we learn the poor actions for greater Q values, the minimum Q value approximately corresponds to the best action in the original environment.

The straightforward approach is using inverse rewards, i.e. negative rewards, to learn the poor actions in the reinforcement learning algorithm. This is our basic idea called *inverse policy*, and the corresponding Q value is,

$$Q_{t+1}^-(s_t, a_t) = Q_t^-(s_t, a_t) + \alpha[-R_t + \gamma \max_{b \in \mathbf{A}} Q_t^-(s_{t+1}, b) - Q_t^-(s_t, a_t)] \tag{5}$$

We will discuss the details in section 2.1 and show that inverse policies are competitive with the original algorithms in section 3. However, they are just the alternative approaches and are not obviously better than the original reinforcement learning algorithms.

Combining the original reinforcement algorithm and its inverse policy, our next approach is to leverage them to correct the mis-actions for each other. In this algorithm, we have two parameterized value functions: $Q(s, a; \theta^+)$ for the original Q values and $Q(s, a; \theta^-)$ for the inverse Q values. We

use the inverse policy to help the original policy to choose the potential next steps, and vice versa on the inverse policy side.

Eventually, we have the following *hybrid policy*. The policy chooses the action with the maximum $Q^H$ value at each step.

$$Q^H(s_t, a_t) = C(Q^+(s_t, a_t), Q^-(s_t, a_t)) \approx C(Q(s_t, a_t; \theta_t^+), Q(s_t, a_t; \theta_t^-)), \quad (6)$$

where $Q^+$ is the Q value for the original policy, $Q^-$ is the Q value for the inverse policy and $C(q_1, q_2)$ is a simple function that merges two values into one.

Note that the above improvement using hybrid policy is used for off-policy algorithms. For on-policy algorithms like SARSA and on-policy actor-critic, the similar improvement is simpler. We use $Q^+$ and $Q^-$ functions to approximate the Q value along the same trajectories. Then similar $Q^H$ is used to merge the values in order to make final decision. We will discuss this issue in section 2.3 and take on-policy actor-critic as the example.

## 1.3 CONTRIBUTIONS

We make the following contributions in this paper.

1. To the best of our knowledge, we are the first one who use inverse rewards for reinforcement learning and prove the convergence of the inverse policies. The policies with the inverse rewards are competitive with those policies with the original rewards.

2. We merge the original policies and the inverse policies as the hybrid policies. In the hybrid policies, the inverse policies effectively correct many mis-actions generated by the original policies, and vice versa. (See section 3.2)

3. Both the inverse policy and the hybrid policy are broad-spectrum for both on-policy and off-policy reinforcement learning algorithms. We have applied them on deep Q-learning, double Q-learning, and on-policy actor-critic. They are also potentially applicable for the policy gradient-based algorithms and other cutting-edge algorithms.

4. Our evaluation shows that the hybrid policies are much better than the original policies. In the experiments, the hybrid polices gain up to 63.8%, 97.8%, and 54.7% rewards more than the original deep Q-learning, double Q-learning and on-policy actor-critic for some OpenAI gym environments, respectively. Moreover, our improvements are much more stable.

## 2 INVERSE POLICY AND HYBRID POLICY

### 2.1 INVERSE POLICY

The basic idea of inverse policy in section 1.2 transforms the reward $r$ to $-r$ and uses $-r$ for learning. Recall equation 5. In practice, we use $Q(s, a; \theta^-)$ to approach $Q^-(s, a)$:

$$\begin{aligned} Q_{t+1}^-(s_t, a_t) &\approx Q(s_t, a_t; \theta_{t+1}^-) \\ &= Q(s_t, a_t; \theta_t^-) + \alpha^-[-R_t + \gamma^- \max_{b \in \mathbf{A}} Q(s_{t+1}, b; \theta_t^-) - Q(s_t, a_t; \theta_t^-)]. \end{aligned} \quad (7)$$

And the updating strategy is,

$$\theta_{t+1}^- = \theta_t^- + \alpha^-(Y_t^{Q^-} - Q(s_t, a_t; \theta_t^-))\nabla_{\theta_t^-} Q(s_t, a_t; \theta_t^-) \quad (8)$$

$$Y_t^{Q^-} = -R_t + \gamma^- \max_{b \in \mathbf{A}} Q(s_{t+1}, b; \theta_t^-). \quad (9)$$

Like the original deep Q-learning algorithm Melo, the updating strategy in equation 8 and 9 converges to $Q^{-*}$ (proved in Appendix A). It follows that the optimal inverse policy can be obtained as:

$$\forall s \in \mathbf{S}, \pi^{-*}(s) = \arg\max_{a \in \mathbf{A}}(-Q^{-*}(s, a)) = \arg\min_{a \in \mathbf{A}} Q^{-*}(s, a). \quad (10)$$

Note that the inverse policy is a *broad-spectrum method* for many reinforcement learning algorithms. For example, we can use the following equations for double Q-learning,

$$Y_t^{DoubleQ-} = -R_t + \gamma^- Q(s_{t+1}, \arg\max_{b \in \mathbf{A}} Q(s_{t+1}, b; \theta_t^-); \theta_t^{-'}) \quad (11)$$

$$\theta_{t+1}^- = \theta_t^- + \alpha^-(Y_t^{DoubleQ-} - Q(s_t, a_t; \theta_t^-))\nabla_{\theta_t^-} Q(s_t, a_t; \theta_t^-). \quad (12)$$

Similarly, the inverse policy for on-policy algorithms is much simpler as follows.
$$\theta_{t+1}^- = \theta_t^- + \alpha^-(-R_t + \gamma^- Q(s_{t+1}, a_{t+1}; \theta_t^-) - Q(s_t, a_t; \theta_t^-))\nabla_{\theta_t^-} Q(s_t, a_t; \theta_t^-). \tag{13}$$

## 2.2 HYBRID POLICY

We continue from the inverse policy to introduce the *hybrid policy*, which combines the original and inverse policy to get a better one. Recall that one major problem of deep Q-learning is the policy-degradation caused by the wrongly-estimated Q values in the learning process. The traditional Q-learning is based on an assumption that the target Q value, e.g. $Y^Q$ as shown in equation 3, is a good approximation of $Q^*$, by following the operator $max$ to find the best action in the next state in the current Q function. However, in some cases sub-optimal actions are selected and result in mis-estimated Q values. This kind of error is intrinsic and has been discussed in the work of Thrun & Schwartz (1993)van Hasselt (2011).

In this paper, we suggest that the combination of the original deep Q-learning policy and the inverse policy leads to a better one, i.e. selects better actions than each single one of the former both, since some wrongly-selected actions of one Q function can be corrected by the other one.

Following this feature, the general idea of hybrid deep Q-learning algorithm is to combine the original and the inverse Q values into *hybrid Q value*, which can help select the optimal actions for both the learning and the policy.

The novel algorithm includes two sets of parameterized Q value functions, denoted as $Q(\theta^+)$ (the original Q function) and $Q(\theta^-)$ (the inverse Q function). For each function, we compute its hybrid Q value $Q^H$, which is influenced by its counterpart Q function,
$$Q^{+H}(s, a) = C(Q(s, a; \theta^+), Q(s, a; \theta^-)), \tag{14}$$
$$Q^{-H}(s, a) = C(Q(s, a; \theta^-), Q(s, a; \theta^+)). \tag{15}$$
The operator $C(q_1, q_2)$ [1] is a simple function, which uses the value of $q_2$ to adjust the value of $q_1$.

For both $Q^+$ and $Q^-$ functions, $Q^H$ is applied to determine the optimal action, both in the sampling and in the updating rule. During the sampling and exploration, $Q^+$ and $Q^-$ pick their actions by maximizing $Q^H$ values, as the following policies,
$$\forall s \in \mathbf{S}, \pi^{+H}(s) = \arg\max_{a \in \mathbf{A}} Q^{+H}(s, a), \pi^{-H}(s) = \arg\max_{a \in \mathbf{A}} Q^{-H}(s, a). \tag{16}$$
Meanwhile, in the updating rule of deep Q-learning, $Q^H$ is applied to select the optimal action. The computation of target Q value, $Y_t^Q$, should be re-written for both Q functions,
$$Y_t^{Q+} = R_t + \gamma^+ Q(s_{t+1}, \arg\max_{b \in \mathbf{A}} Q_t^{+H}(s_{t+1}, b); \theta_t^+),$$
$$Y_t^{Q-} = -R_t + \gamma^- Q(s_{t+1}, \arg\max_{b \in \mathbf{A}} Q_t^{-H}(s_{t+1}, b); \theta_t^-), \tag{17}$$
while the gradient descent method used in Q value approximation as the traditional deep Q-learning remains unchanged. Note that $Q^H$ influences only the choice of optimal action, not the Q value itself.

In equation 16 and 17, $Q^+$ function chooses the optimal action depending on not only $Q(s, a; \theta^+)$ but also $Q(s, a; \theta^-)$. It means that the inverse policy is helping the original policy to choose potential next steps. Symmetrically, the original policy also helps correcting the inverse policy.

Eventually, we have the following *hybrid policy* $\pi^H$, which is the overall policy of our agent. In our algorithm, $\pi^H$ is the same as $\pi^{+H}$.
$$\pi^H(s) \simeq \pi^{+H}(s) = \arg\max_{a \in \mathbf{A}} C(Q(s, a; \theta^+), Q(s, a; \theta^-)). \tag{18}$$
An optimal policy $\pi^{H*}$ can be obtained upon the convergence of $Q^+$ and $Q^-$, which is,
$$\forall s \in \mathbf{S}, \pi^{H*}(x) = \arg\max_{a \in \mathbf{A}} C(Q^{+*}, Q^{-*}). \tag{19}$$

During the learning process, $Q^+$ and $Q^-$ are equally important as they both contribute to the hybrid Q value ($Q^H$). In practice, both functions are trained simultaneously, with separately sampling and learning process ,so that each one improves itself and helps the other at the same time.

---

[1] In our experiment, the combination operator $C$ is defined as $C = \lambda q_1 + (1 - \lambda)(-q2)$, where $\lambda$ is an arbitrary interpolation factor. Note that $C(q_1, q_2) \neq C(q_2, q_1)$.

The hybrid Q-learning algorithm can be easily generalized in other value-function-based algorithms. In double Q-learning, the $Q^H$ value can be computed to improve the original algorithm by using the following equations,

$$Y_t^{DoubleQ+} = R_t + \gamma^+ Q(s_{t+1}, \arg\max_{a \in \mathbf{A}} C(Q(s_{t+1}, a; \theta_t^+), Q(s_{t+1}, a; \theta_t^-)); \theta_t^{+'}), \tag{20}$$

$$Y_t^{DoubleQ-} = -R_t + \gamma^- Q(s_{t+1}, \arg\max_{a \in \mathbf{A}} C(Q(s_{t+1}, a; \theta_t^-), Q(s_{t+1}, a; \theta_t^+)); \theta_t^{-'}). \tag{21}$$

As we discussed in section 1, the hybrid policy for on-policy algorithms is simpler. We implement the following on-policy $Q^H$ using the original $Q(s, a; \theta^+)$ and $Q(s, a; \theta^-)$ in equation 13,

$$Q^H(s, a) = C(Q(s, a; \theta^+), Q(s, a; \theta^-)). \tag{22}$$

### 2.3 PRACTICE ON ON-POLICY ACTOR-CRITIC ALGORITHMS

We manage to port our algorithm of hybrid policy to the on-policy actor-critic (AC) algorithm, which is widely used in the current reinforcement learning cases. There exist several variants of AC algorithm, such as DDPG David Silver (2014), A3C Volodymyr Mnih (2016) and DPPO Schulman et al. (2017). We share our experience on the original on-policy AC algorithm, and the practice should be reproducible on other AC-based algorithms.

AC uses value-function-based critic to learn the values of the current policy (i.e. $Q^\pi$) and guides the actor's policy update. The idea of hybrid policy is used to build a more stable and accurate value function for the critic. We choose on-policy AC to show that our approach is also effective for on-policy algorithms. Like in deep Q-learning, an inverse-value function is created, denoted as the *inverse critic*. This critic takes reversed rewards and is trained simultaneously with the native critic. Then the two critics evaluate the behavior of the actor together. We define the target value to be approximated by both native and inverse critics as,

$$\theta_{t+1}^{Q^+} = \theta_t^{Q^+} - \alpha^+ (R_t + \gamma^+ Q^\pi(s_{t+1}, a_{t+1}; \theta_t^{Q^+}) - Q^\pi(s_t, a_t; \theta_t^{Q^+})) \nabla_{\theta^{Q+}} Q^\pi(s_t, a_t; \theta_t^{Q^+}),$$

$$\theta_{t+1}^{Q^-} = \theta_t^{Q^-} - \alpha^- (-R_t + \gamma^- Q^\pi(s_{t+1}, a_{t+1}; \theta_t^{Q^-}) - Q^\pi(s_t, a_t; \theta_t^{Q^-})) \nabla_{\theta^{Q-}} Q^\pi(s_t, a_t; \theta_t^{Q^-}),$$

$$\theta_{t+1}^\pi = \theta_t^\pi - \beta \nabla_{\theta^\pi} \log \pi(a_t | s_t; \theta_t^\pi) C(Q^\pi(s_t, a_t; \theta_t^{Q^+}), Q^\pi(s_t, a_t; \theta_t^{Q^-})).$$

$$\tag{23}$$

where $\beta$ is the actor's learning rate, $\theta^{Q^+}$ and $\theta^{Q^-}$ are the parameters of the original and inverse critics, and $\theta^\pi$ is the parameter of the actor. We presents the experiment results based on this approach in Section 3 and show that our approach can learn the value of $Q^\pi$ more efficiently and stably.

## 3 EVALUATION

### 3.1 OVERVIEW

We use CartPole ope (a), Mountain Car ope (c), and Pendulum ope (d) in OpenAI gym v0.9.6 ope (b) without any modification. CartPole is run with 4000 episodes and 200 maximum steps per episode. Mountain Car is run with 2000 episodes and 1000 maximum steps. [2] Pendulum is run with 1000 episodes and 2000 maximum steps. We ran the experiments on an Intel x86 machine on Ubuntu Linux 16.4 with TensorFlow 1.6 ten; Abadi et al. (2016) as our RL platform. Each algorithm is implemented as a neural network, which consists of one fully-connected layer with 128 hidden-nodes. For each game, we test the original method of deep Q-learning, double Q-learning, and on-policy actor-critic [3] as our baseline firstly. Then we apply inverse policy and hybrid policy on these algorithms.

Note that we use the off-line training in this paper. We train the policy for an entire episode and use the trained policy to test the benchmark for an entire episode. The police is unchanged during the

---

[2] In Mountain Car, we use $|nextState[0] - currentState[0]|$ as the reward for training because the original deep Q-learning and double Q-learning obtain very poor results using the original reward generated by the environment, which is meaningless. Nevertheless, we use the original rewards of Mountain Car in figure 2 and table 2 to make the results easy to understand.

[3] Because deep Q-learning and double Q-learning cannot process the continuous action space, we discretize the continuous action space into 5 discrete actions for Pendulum. We still use continuous action space in actor-critic algorithm for Pendulum.

Table 1: The algorithm parameters.

| | Cart Pole | | | Mountain Car | | | Pendulum | | |
|---|---|---|---|---|---|---|---|---|---|
| | $\alpha$ | $\gamma$ | $C(q_1, q_2)$ | $\alpha$ | $\gamma$ | $C(q_1, q_2)$ | $\alpha$ | $\gamma$ | $C(q_1, q_2)$ |
| Deep Q[+] | 0.002 | 0.9 | $C\|_{\lambda=0.5}^2$ | 0.001 | 0.9 | $C\|_{\lambda=0.5}^2$ | 0.001 | 0.9 | $C\|_{\lambda=0.5}^2$ |
| Deep Q[-] | 0.002 | 0.7 | | 0.001 | 0.9 | | 0.00005 | 0.9 | |
| Double Q[+] | 0.002 | 0.9 | $C\|_{\lambda=0.5}^2$ | 0.001 | 0.9 | $C\|_{\lambda=0.5}^2$ | 0.001 | 0.9 | $C\|_{\lambda=0.5}^2$ |
| Double Q[-] | 0.002 | 0.9 | | 0.001 | 0.9 | | 0.00005 | 0.9 | |
| AC(Critic[+])[1] | 0.001 | 0.9 | $C\|_{\lambda=0.5}^2$ | 0.005 | 0.9 | $C\|_{\lambda=0.5}^2$ | 0.001 | 0.9 | $C\|_{\lambda=0.5}^2$ |
| AC(Critic[-])[1] | 0.0005 | 0.7 | | 0.005 | 0.9 | | 0.001 | 0.9 | |

+ The parameters for the original policy.
- The parameters for the inverse policy.
1 For the actor, $\alpha$ is 0.00002(CartPole), 0.0005(Mountain Car) and 0.0001(Pendulum), respectively.
2 The combination function $C(q_1, q_2) = \lambda q_1 + (1 - \lambda)(-q_2)$.

testing episode. In comparison, many other works employ on-line training which trains and tests the policy in the same episode. We choose the off-line training because it is more lightweight and efficient in practice, and easier to expose the unstability of the policies. The performance and the stability of the off-line training is obviously worse than that of the on-line training.

Some hyper-parameters that are set in the experiments include: (1) $\alpha$, as the *learning rate* for the gradient descent method, (2) $\gamma$, as the *discount factor* in the accumulation of Q values, (3) $\lambda$, as the factor used in the combination operator $C(q_1, q_2)$, which controls the merging of Q values. For each experiment, these parameters are separately adjusted to achieve the best outcome in table 1.

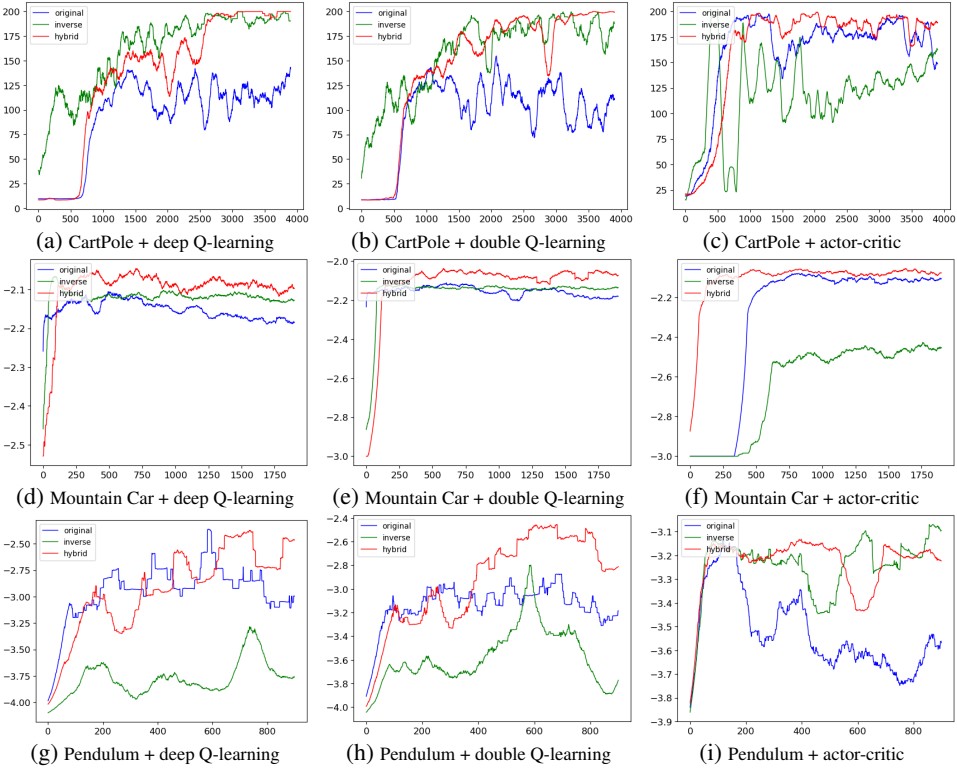

(a) CartPole + deep Q-learning   (b) CartPole + double Q-learning   (c) CartPole + actor-critic

(d) Mountain Car + deep Q-learning   (e) Mountain Car + double Q-learning   (f) Mountain Car + actor-critic

(g) Pendulum + deep Q-learning   (h) Pendulum + double Q-learning   (i) Pendulum + actor-critic

Figure 2: The results of Q-learning, double Q-learning and on-policy actor-critic algorithms, on games of CartPole, Mountain Car and Pendulum. The comparison of algorithms in three variants: the traditional, the inverse policy and the hybrid policy. The x-axis is the iterations of episodes and y-axis is the **100-step average rewards** around the corresponding episode. In game Mountain Car and Pendulum, we use **log scaled y-axis**, which is computed as $y' = -log(-y)$.

Figure 2 demonstrates the combined results of the different policies, for each algorithm and each game. We compute the average reward of the 100 episodes around each episode, so that we can

Table 2: The mean and standard deviation of rewards for the final on-line performance.

| | Cart Pole | | Mountain Car | | Pendulum | |
|---|---|---|---|---|---|---|
| Algorithm | mean | std dev. | mean | std dev. | mean | std dev. |
| Deep Q Original | 122.71 | 54.94 | -151.02 | 26.59 | -990.08 | 3238.70 |
| Deep Q Inverse | 194.87 | 26.33 | -134.78 | 17.44$^\dagger$ | -5647.43 | 4160.85 |
| Deep Q Hybrid | 198.58$^\dagger$/52.2%$^\ddagger$ | 16.33$^\dagger$ | -127.19$^\dagger$/15.8%$^\ddagger$ | 31.86 | -358.79$^\dagger$/63.8%$^\ddagger$ | 695.38$^\dagger$ |
| Double Q Original | 100.75 | 55.78 | -153.94 | 24.84 | -1537.95 | 4076.06 |
| Double Q Inverse | 181.44 | 52.48 | -138.25 | 18.91$^\dagger$ | -5821.21 | 4889.33 |
| Double Q Hybrid | 199.35$^\dagger$/97.8%$^\ddagger$ | 4.07$^\dagger$ | -116.6$^\dagger$/24.1%$^\ddagger$ | 23.70 | -649.91$^\dagger$/58.1%$^\ddagger$ | 1450.11$^\dagger$ |
| Actor-Critic Original | 169.19 | 36.54 | -128.22 | 32.64 | -3696.94 | 4442.36 |
| Actor-Critic Inverse | 148.42 | 28.74 | -282.98 | 99.14 | -1246.66$^\dagger$ | 532.72$^\dagger$ |
| Actor-Critic Hybrid | 188.59$^\dagger$/11.9%$^\ddagger$ | 26.01$^\dagger$ | -120.0$^\dagger$/6.4%$^\ddagger$ | 27.09$^\dagger$ | -1677.30 /54.7%$^\ddagger$ | 751.33 |

$\dagger$ the best result (i.e. higher mean value and lower standard deviation) in the comparable experiments.
$\ddagger$ the improvement rate of hybrid policy compared with the original algorithm.

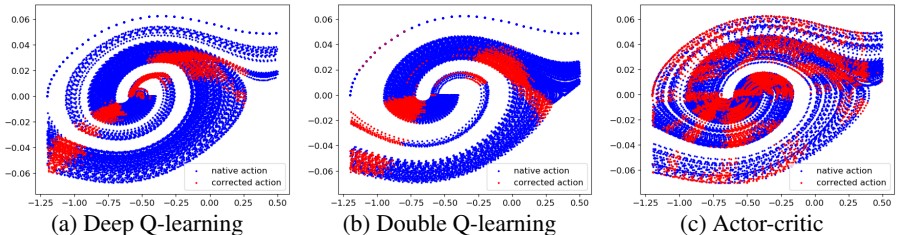

|      (a) Deep Q-learning      |      (b) Double Q-learning      |      (c) Actor-critic      |

Figure 3: The correction of policies on Mountain Car experiment, with deep Q-learning, double Q-learning and actor-critic. x-axis and y-axis are the the velocity and the position of the car, respectively.

observe the change of expected reward along the training. Note that in the experiment of Mountain Car and Pendulum, the result range is transformed to log scale for ease of review. The detailed figures showing the rewards at each episode can be found at Appendix B.

In CartPole as figure 2a and figure 2b, the inverse policies are better than the original policies, and learn faster. This is because the game has only two actions, and the failures give valuable information to agent, which make it favorable to the inverse policy. Meanwhile, as expected, in both cases the hybrid policies manage to surpass the inverse ones with more steady increase in the end. The inverse policy in figure 2c for actor-critic is less impressive, which matches the original in general, with a better stability. Still, the hybrid policy is the best.

We can get the similar conclusion for Mountain Car from figure 2d 2e 2f, where the hybrid policy outperforms the others at end of training. Though for actor-critic in figure 2f the hybrid policy is only slightly better than the original in the end, it learns the suitable policy much earlier than the others.

For Pendulum (Fig 2g2h 2i), we deliberately set a large amount of training steps to observe the policy degradation and stability. We can see the obvious degradation for all three algorithms and for the inverse policy in double Q-learning. In comparison, the hybrid policy can effectively mitigate the trend of policy degradation in a long and intensive training process. Note that for actor-critic, the hybrid policy works well on continuous action space, which shows the generalization of our approach.

In table 2, we evaluate the performance by computing the mean and standard deviation with respect to *the final on-line performance* van Hasselt (2011), i.e. the episode scores obtained in the last 10% of the whole training runs. The mean shows the score expectation by the concluded policy, and the standard deviation measures the stability. With no surprise, in almost all cases the hybrid policy gives the best average reward. In most cases, hybrid policy is also the most stable policy, except in a few cases inverse policy behaves more steadily. Meanwhile, either one of the two variant policies is better than their original peer in almost all cases.

## 3.2 DISCUSSIONS AND INSIGHTS

Some insights can be obtained by studying the performance of inverse policy. From our experience, inverse policy learns directly from the bad experience of agent, which makes it quite effective when useful information is only received at the failure of the game. Note that such failure-sensitive problem

is quite common in sparse-reward problems. As in CartPole, inverse agent can learn effective policy much more quickly, based on the massive failure experience from the beginning episodes. Therefore, in failure-sensitive problems, inverse policy can be used to boost the policy in short time.

We found that, for inverse policies, exploration is quite crucial. Since the inverse agent picks the bad move that leads to a fast failure, it is easy to be stuck at local optima. The way we choose to address this problem by adding large random noise to the selection of inverse agent, such as a high random factor $\epsilon$ in the inverse Q-learning algorithm.

We also studied how the policy of the original is influenced by the $Q^-$ function. Mountain Car includes two continuous state dimensions: the position ([-1.2,0.6]) and the velocity ([-0.07, 0.07]) of the car. We record all the states and chosen actions in the final 10% of training. Note that these actions are actually chosen by the hybrid policy. At each step, we also record the actions that $Q^+$ policy thought as optimal. Then we show the states where $Q^+$ policy were corrected by the $Q^-$ function in figure 3. For actor-critic, we record the actions where the original critic and inverse critic have different evaluation on the chosen action (with respect to which action has the max value).

In both figures of figure 3a and figure 3b, most corrected actions gather at the states of low or medium car velocity [-0.03,0.03]. In these states the agent is more frequently trained. According to the study of Q-learning overestimation Thrun & Schwartz (1993)van Hasselt (2011), biased noise is cumulative, which worsens with more samples. Plus, since the following rewards at these states are close, the action value gap is relatively narrow. So sub-optimal actions value is more likely to surpass the true optimal action Bellemare et al. (2016). By correcting these mis-actions, $Q^-$ policy effectively improve the agent's behavior. From the result of actor-critic in figure 3c, for on-policy Q functions, the mis-estimation of Q values are more like unbiased noise, since the corrected actions scatters over states. This result matches the analysis in van Hasselt (2011).

## 4 RELATED WORKS

Many studies are performed to solve the problem of sub-optimal policy and the policy-degradation. All value-function-based algorithms may benefit from the mitigation of mis-estimated Q values. Some algorithms, such as double Q-learning van Hasselt et al. (2016) and dueling Q-learning Wang et al. (2016), proposed novel Q-learning architecture to obtain better approximation of $Q^*$ and more stable learning process. In the work of Bellemare et al. (2016), this problem is addressed in a more fundamental way. The paper proposed a new *consistent Bellman operator* in the Q-learning update rule, which is claimed to be able to increase the gap of action values in every state, while maintaining the optimal policy. It is shown that by modifying the Bellman operator, a more stable policy can be achieved. In our work, the same insight is given.

The definition and usage of reward is another key point. Several studies focus on the idea of leveraging different forms of reward. Starting from the Horde architecture Sutton et al. (2011), the concept of *pseudo-reward function* is proposed to represent any useful feature-based signals. Based on those rewards, separate *general value functions (GVF)* are trained in separate agents. Eventually, multiple agents are combined together to solve complex problems. This idea continues in later works of UNREAL Jaderberg et al. (2016) and HRA Van Seijen et al. (2017). In these studies, one complex task is studied by breaking it down into several reward functions, or by associating it with axillary tasks aside. The difference between the works above and ours is that, we have shown that an impressive improvement can be achieved by just leveraging the inverse reward, instead of using additional reward functions and agents.

## 5 CONCLUSION

The inverse policies and hybrid policies in this paper improve the original reinforcement learning algorithms significantly. We obtained 63.8%, 97.8%, and 54.7% more rewards than the original deep Q-learning, double Q-learning, and on-policy actor-critic algorithms do for some OpenAI gym environments. More importantly, our new policies are much more stable in the experiments.

We will apply our new algorithms on more complicated environments. Although our improvements may be easily applied on the algorithms based on policy gradient intuitively, we have to write the theory for them and run some solid tests in the near future.

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
