# OpenReview forum: "Hybrid Policies Using Inverse Rewards for Reinforcement Learning"
_ICLR.cc/2019/Conference_

### Official Review · AnonReviewer2 · 2018-11-02
**Interesting idea that may only work in some specific cases**

**Rating:** 4
**Confidence:** 5

**Review:**

Given an MDP <S, A, T, R>, the paper suggests to learn both the optimal Q function of that MDP (denoted Q^+), but also that of the MDP <S, A, T, -R> (denoted Q^-). The basic idea is that min_a Q^-(s, a) could be a good action for the initial MDP. Based on this idea, the authors propose to combine Q^+ and Q^- with a linear combination in order to obtain what they call a hybrid policy.

The proposed idea is indeed interesting and I find the experimental results surprising. It is not clear to me why the policy obtained from Q^- does better than Q^+. Theoretically, this should not happen: if we have the exact optimal Q function, Q^-, for <S, A, T, -R>, the policy defined by argmin_a Q^-(s, a) in every state s may be suboptimal in <S, A, T, R>. Is there a good conjecture/explanation for why the policy induced by Q^- works so well in 2(a) and (b)?

The authors chose to report the results using off-line training, which seem to favor their proposition. What are the results for on-line training?

In the experimental part, I think the authors should also report the results of the method that consists in learning two Q^+ and combining them with an average. This baseline would help understand if the good performance of hybrid policies really comes from learning Q^-.

Obtaining hybrid policies faces one important issue, which is the need to perform two actions in the environment in a given state, one for Q^+ and the other for Q^-. Therefore, the proposition seems to be doable only when one has access to a simulator.

The writing is generally clear, but the paper should be checked for typos.

---

### Official Review · AnonReviewer3 · 2018-11-02
**Proposes enhancing reinforcement learning by also solving the same MDP but with the negative of the original rewards.**

**Rating:** 2
**Confidence:** 5

**Review:**

This paper proposes that, in addition to learning the normal action values, an RL agent should also learn the action values for an alternate “inverse” problem consisting of the same transitions as the original MDP and the negative of the original rewards. The intuitive argument is that the values for the inverse problem clearly identify what actions should not be taken. Results are presented on OpenAI Gym problems in which the new method performed better than conventional methods.

The paper is not yet ready for publication for many reasons. First, the idea is not presented clearly, and it is not clear why it ever could be sensible. The inverse problem has a different solution than the base problem. Its solution would appear to have an arbitrary relationship to the base problem’s solution. The two optimal policies may choose different actions, as suggested in the text, but this is not necessarily true; in some states the two policies may choose the same action. I don’t see how anything can be said in general, and no significant theoretical results are presented. (They do prove a form of convergence on the inverse problem, but this is not a new result; the inverse problem is just another problem and needs no new result.)

The new hybrid method is never fully explained (e.g., the reader has to guess at what Q^H is). But by combining the solutions to the base and inverse problems in some way, it seems inevitable that the final optimal policy would be changed. Suppose the function approximation is completely successful and the correct values are exactly found. Then those for the base problem would give the optimal policy. Any alteration of them by the correct action-value function for the inverse problem could only make them worse and could only cause them to produce worse (or the same) behavior. There is no room for improvement and so this technique could only make things worse asymptotically if the function approximation is completely successful. This is the only thing that I see that can be clearly said about the new technique, and of course it is not a good thing.

There are results presented, but they are not well done; they provide no significant evidence for any conclusion. The methods are not completely presented, and the results seem to be for a single run, in which case any relative ordering of the methods could be obtained.

Generally, the paper is unfortunately poorly written. The grammar is not good. The notation is unnecessarily complex and confusing. The citations are made in an unusual, poor way.

---

### Official Review · AnonReviewer4 · 2018-11-12
**Interesting idea, but requires more rigorous evaluation**

**Rating:** 3
**Confidence:** 4

**Review:**

The paper proposes a method for improving the stability of reinforcement learning with value function approximation, e.g., deep Q-learning. The key idea is fitting a Q function to rewards, fitting another Q function to negative rewards, then estimating Q values using a linear combination of the two Q functions. The method is applied to DQN, double DQN, and on-policy actor-critic on the CartPole, Mountain Car, and Pendulum tasks in OpenAI Gym.

The writing isn't clear, especially in the introduction. Phrases like "risky", "badness of a state", and "inverse policy" are used without definition.

The experiments only test one value of \lambda. Since \lambda is the one hyperparameter that controls the degree to which the inverse rewards influence the Q value estimates, I think it is critical to test the performance of the proposed method under various values of \lambda (e.g., by sweeping the unit interval in increments of 0.1).

One of the central claims is that the proposed Q value estimator gives more accurate estimates of returns than the estimators used in previous deep Q-learning methods. However, the experiments never compare the predicted Q values to the true values, as is done in [3].

The experiments only evaluate the proposed method on the CartPole, Mountain Car, and Pendulum tasks, which have very small action spaces. I suspect the benefit of the proposed method will be smaller in environments with a larger number of possible actions, since the inverse policy may fail to accurately estimate the values of actions that are neither the best nor the worst at any given time.

One of the central claims is that the proposed method improves the stability of Q-learning, but it is unclear how many random seeds were tested in Figure 2 and Table 2. It appears that only the data from one training run was used, and the reported standard deviations are computed using the last 10% of episodes in that single training run. Furthermore, the curves are smoothed using a moving average with a window size of 100 episodes. Together, these two details make it extremely difficult to evaluate the claim that the proposed algorithm is more stable, and also makes it difficult to evaluate the significance of the differences between the method's performance and the baselines. [1] shows how results on a small number of random seeds tend to not be reproducible.

[1] https://arxiv.org/pdf/1709.06560.pdf
[2] http://www.gatsby.ucl.ac.uk/~dayan/papers/cjch.pdf
[3] https://arxiv.org/pdf/1509.06461.pdf

---

### Meta-Review · Area_Chair1 · 2018-12-14

**Confidence:** 5
**Recommendation:** Reject

**Metareview:**

Pros:
- an original idea: learn an additional inverse policy (that minimizes reward) to help find actions that should be avoided.

Cons:
- not clearly presented
- conclusions are not not validated
- empirical evidence is weak
- no rebuttal

The three reviewers reached consensus that the paper should be rejected in its current form, but make numerous suggestions for improving it for a future submission.